# Land Management Contributes significantly to observed Vegetation Browning in Syria during 2001–2018

Tiexi Chen[1,2], Renjie Guo[1], Qingyun Yan[3], Xin Chen[1], Shengjie Zhou[1], Chuanzhuang Liang[1], Xueqiong Wei[1], Han Dolman[4]

[1]School of Geographical Sciences, Nanjing University of Information Science and Technology, Nanjing 210044, China
[2]School of Geographical Sciences, Qinghai Normal University, Xining 810008, China
[3]School of Remote Sensing and Geomatics Engineering, Nanjing University of Information Science and Technology, Nanjing 210044, China
[4]Department of Earth Sciences, Vrije Universiteit Amsterdam, Amsterdam 1081 HV, The Netherlands

*Correspondence to*: Tiexi Chen (txchen@nuist.edu.cn)

**Abstract.** Climate change and human activities have significant impacts on terrestrial vegetation. Syria is a typical arid region with a water-limited ecosystem and has experienced severe social unrest over the last decades. In this study, changes in vegetation and potential drivers in Syria are investigated. By using an enhanced vegetation index (EVI), a general browning trend is found in Syria during 2001–2018 with the EVI decreasing at a rate of $-0.8 \times 10^{-3}$ $yr^{-1}$ ($p < 0.1$). The decrease of the EVI is mainly found in the north region, whereas the west region still maintains an increasing trend. The residual analysis indicates that besides precipitation, human activities also contribute significantly to the EVI decrease, which is confirmed by the decrease in rainfall use efficiency. Moreover, a Paired Land Use Experiment (PLUE) analysis is carried out in the Khabur River Basin where croplands are widely distributed in adjacent regions of Syria and Turkey. The time series of the EVIs over these two regions are highly correlated ($r = 0.8027$, $p < 0.001$), indicating that both regions are affected by similar climate forcing. However, vegetation in Syria and Turkey illustrates contrary browning ($-3 \times 10^{-3}$ $yr^{-1}$, $p < 0.01$) and greening trends ($4.5 \times 10^{-3}$ $yr^{-1}$, $p < 0.01$), respectively. Relative reports have reported that social unrest had induced insufficient irrigation and lack of seeds, fertilizers, pesticides and field managements. Therefore, we concluded that the decline of vegetation in the north Syria is driven by the change of land managements.

## 1. Introduction

Vegetation is a key component of the Earth system and plays an important role in the water and energy cycle. It is also one of the basic natural conditions for human survival. Therefore, the variations, driving mechanisms and impacts of the vegetation have been widely studied (Li et al., 2018; Piao et al., 2020; Zhu et al., 2016). Globally, observations have shown a general greening trend, although browning, i.e. loss of vegetation activity occurred in some regions (Jong et al., 2012; Zhang et al., 2017). Here, the greening (browning) usually refers to changes in vegetation conditions indicated by the positive (negative) long-term trend of vegetation index (Piao et al., 2020).

Recent advances in remote sensing technology and the establishment of long-term vegetation indexes have greatly facilitated the research on the long-term monitoring of large-scale vegetation. Indexes such as the normalized difference vegetation index (NDVI) developed by the Global Inventory Monitoring and Modeling Studies (GIMMS), the enhanced vegetation index (EVI) product based on the Moderate Resolution Imaging Spectroradiometer (MODIS) and the leaf area index (LAI) derived from remote sensing data are widely used in the study of long-term vegetation changes (Forzieri et al., 2017; Ju and Masek, 2016).

The main climatic factors restricting vegetation include temperature, precipitation, radiation, and the fertilization effect of $CO_2$ (Chen et al., 2020; Keenan and Riley, 2018; Schimel et al., 2015). At the global scale, the long-term trends of these driving factors manifest itself as warming and increasing atmospheric $CO_2$ concentration. Model results show that the $CO_2$ fertilization effect on global scale greening contributes over a half (Zhu et al., 2016). However, the vegetation index derived from remote sensing data shows that the proportion of global cropland greening is relatively high, which is different from model simulations on the $CO_2$ fertilization effect. At the same time, the agricultural modernization is also considered to have a great impact on vegetation growth (Chen et al., 2019).

At present, the influences of climate change on vegetation have been widely simulated by models, but the model performance on the variation of land-use types is relatively poor, and land management practices are not well represented (Pongratz et al., 2014; Prestele et al., 2017). There are two main reasons for these disadvantages. On one hand, the land management data are not complete. Land management involves many aspects, including ecological engineering, economic management of natural vegetation, as well as the seed-selection, irrigation, fertilization and pesticides for croplands. This results in the lack of high-quality datasets with spatio-temporal continuity for land management. On the other hand, it is more difficult to quantify the characteristics of various types of land management. Therefore, a common practice in previous studies is to make a general distinction between natural and human drivers. As a result, it is rather difficult to study the specific management processes in depth (d'Amour et al., 2017; Erb et al., 2017; Meyfroidt et al., 2013).

Social unrest is still a global problem, which is bound to affect regional development and land management activities. However, social unrest has been rarely mentioned in the study of land use and management. Based on the Armed Conflict Location and Event Data Project (ACLED) records (Raleigh et al., 2010), in the past 20 years, there are about 20 countries that have experienced more than 1000 battles. Several countries have involved in national wars, such as (not limited to) Syria, Libya,

Afghanistan, Iraq, Yemen. Therefore, the affected population and land area are at least in the order of 100 million and tens of millions of square kilometers, respectively.

From a land management perspective, Syria deserves great concern. Syria is a typical arid area, the social unrest in Syria is bound to lead to a series of changes in land management, especially the serious destruction to agricultural input caused by the war, which caused severe food security problem (Li et al., 2022). So far, there have been relatively few studies on land

management changes triggered by social unrest. A typical case in the past is the problem of cropland abandonment in Eastern Europe caused by the collapse of the Soviet Union (Alcantara et al., 2013; Schierhorn et al., 2013). Therefore, it is necessary for Syria to carefully study the impact and mechanism of climate and human activities on vegetation changes. This is also the main research objective of this paper. Specifically, in this paper the basic characteristics of vegetation change in Syria are investigated. Furthermore, the impact of land management on vegetation changes is analysed from multiple ways, so as to

verify the hypothesis that the social unrest will significantly affect the vegetation. Although this mechanism is intuitively obvious, it is not easy to obtain effective and definite quantified conclusions. The remainder of this paper is organized as follows. Section 2 summarizes data and methods. Detailed results are presented in section 3. Discussions and conclusions are presented in section 4 and section 5, respectively.

**2. Data and Methods**

**2.1 Study Area**

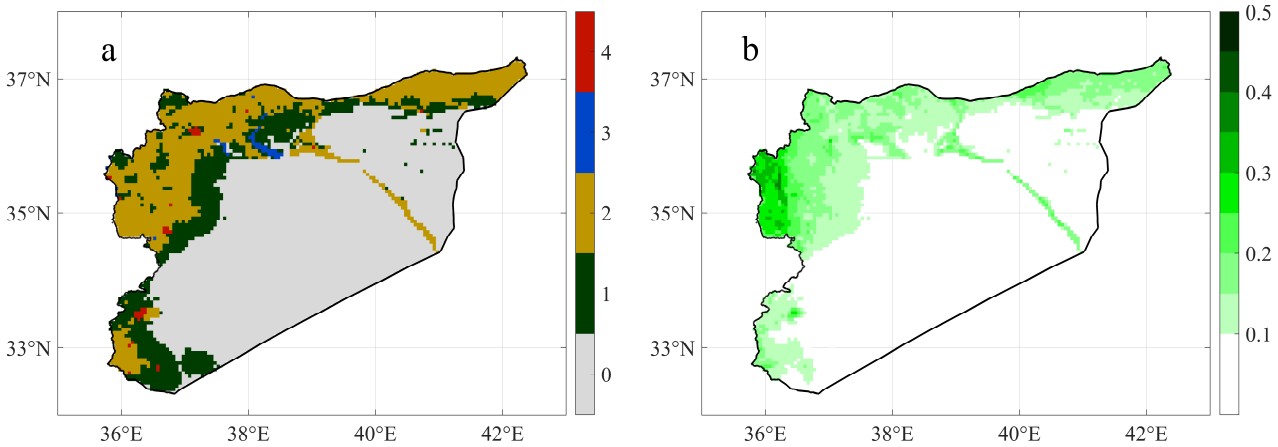

**Figure 1. Land cover and vegetation condition in Syria. (a) Land cover types based on the MOD12C1 data of 2010. Color codes 1–4**
**indicate the natural vegetation, croplands, water bodies and built-up lands, respectively. 0 represents the unused land which is colored in gray. (b) The spatial distribution of annual EVI in Syria.**

Syria (Figure 1) is located in Western Asia, on the east bank of the Mediterranean Sea, bordering Turkey in the north, Iraq in the east, Jordan in the south, and Lebanon and Palestine in the southwest. Syria has a dry climate, and the precipitation is mainly concentrated in the winter half year (November to the next April). The Mediterranean climate in its coastal and northern parts leads to hot, dry summers but mild, rainy winters in these regions. In contrast, the southern region of Syria belongs to tropical desert climate with high temperature and drought throughout the whole year with less precipitation. The main vegetation types in Syria are cropland and grassland, which are primarily distributed in the northern and western coastal areas. The main crop grown in Syria is wheat (Mohammed et al., 2020). Deserts are widely distributed in Syria with sparse vegetation, therefore, only vegetated land where cropland and natural vegetation are selected as the study area (land cover type 1 and 2 in figure 1 (a)).

## 2.2 Land Cover Type Data

In this paper, the land use and land cover dataset of MOD12C1 product in 2010 from MODIS is used, with the spatial and temporal resolutions being 0.05°×0.05° and one year, respectively. This dataset has three different classification schemes. In this study, the IGBP scheme is selected and the data is re-grouped into five categories as shown in Figure 1(a). The natural vegetation includes evergreen needleleaf forests, evergreen broadleaf forests, deciduous needleleaf forests, deciduous broadleaf forests, mixed forests, open shrublands, woody savannas, savannas and grasslands. The second category is cropland that includes croplands and the mosaics of croplands and natural vegetation. The third category is water which contains permanent wetlands and water bodies. The built-up lands remain consistent with the original classification. The rest is classified as the fifth category, unused land, including barren and non-vegetated lands.

## 2.3 Vegetation Index

Optical vegetation indexes are widely used to study the vegetation state. In this paper, the EVI based on MODIS, which can be calculated according to Formula (1), is selected to characterize the vegetation. Compared with the NDVI, the EVI includes the information of the blue band, which can solve the saturation problem of NDVI in densely vegetated areas to a certain extent. The MOD13C2 product is selected, with its spatial resolution (0.05°×0.05°) being consistent with that of the land-use data. The time period of 2001–2018 is selected for the EVI calculation. In October 2018, unusually heavy rain hit Syria and other Middle East regions and caused flooding, which also led to a very high EVI in the following year of 2019. Meanwhile, the heavy rainfall also contributed to desert locust crisis in 2020 (Salih et al., 2020). Because the abnormal value in 2019 has a great interference on trend analysis, it is excluded in this paper. EVI can be calculated as follows:

$$EVI = \frac{G \times (\rho_{NIR} - \rho_{red})}{\rho_{NIR} + C_1 \times \rho_{red} - C_2 \times \rho_{blue} + L} \quad (1)$$

where $\rho_{NIR}$, $\rho_{red}$, $\rho_{blue}$ are the reflectance of the near-infrared, red and blue bands, respectively; $G$ (=2.5) is the gain factor; $C_1$ (=6), $C_2$ (=7.5) and $L$ (=1) are adopted coefficients, respectively.

**2.4 Climatic Factors**

The main climatic factors affecting vegetation changes include radiation, precipitation and temperature. Due to the high temperature and drought in Syria, the moisture content is the main restrictive factor. At the same time, temperature has a huge impact on soil water loss through evaporation. Therefore, precipitation and temperature are selected as environmental factors. The CRU TS4.03 (Brown et al., 2020) dataset with the temporal resolution of one month and spatial resolution of $0.5° \times 0.5°$ is adopted in this paper. Comparatively, the spatial resolutions of temperature and precipitation are much lower than that of vegetation. Due to spatial consistencies of temperature and precipitation over small regions, it is assumed that the vegetation within the 0.5° interval has the same environmental condition.

**2.5 Cropland irrigation data**

Irrigation is an important method for conducting farmland management and increasing production, especially for dryland. Irrigation facilities are also vulnerable to severe social stability and economic fluctuations. We selected the "Global Irrigation District Map" (latest version 5) from FAO (Food and Agriculture Organization of the United Nations), which shows the percentage of the total cropland area used for irrigation around a reference year 2005 with a resolution of 5 minutes (Stefan et al., 2013).

**2.6 Statistical Methods**

In this paper, we applied statistical methods on observations rather than using models. Coherent results of these method are expected to lead a more robust conclusion. The residual trend analysis (Burrell et al., 2017; Evans and Geerken, 2004) is widely used to deduce the impact of human activities by removing climatic factor contributions. Its basic assumption is as follows. Through the regression of vegetation and climatic factors, the trend of residuals is analyzed. For a significant trend of residuals, it is considered that this trend is caused by other driving factors other than climatic factors, which is generally attributed to human activities.

The variation of rainfall use efficiency (RUE) is also analyzed, which is expressed by the ratio of EVI to precipitation (Fensholt and Rasmussen, 2011; Ibrahim et al., 2015). Previous studies have shown that there is a good linear relationship between above ground biomass and accumulated precipitation in semi-arid water-limited ecosystems (Dardel et al., 2014; Fensholt et al., 2013). At the same time, the accumulative remote sensing vegetation index can well represent the aboveground net primary productivity (ANPP). Changes in the RUE may be attributed to the factors other than water conditions, which can be generally considered as the influence of human activities (Leroux et al., 2017). The growing season is defined from February to May (F-M) based on the vegetation phenology (Figure 2).

A Paired Land Use Experiment (PLUE) approach is also applied over two parts of a local region. Both parts have identical climate variations and the difference in vegetation changes of these two parts could be caused by human activities rather than climate drivers.

The Pearson correlation analysis and Mann-Kendall trend analysis are used to analyze the correlation and trends, respectively.

**3. Results**

 **3.1 Seasonal Cycle and Long-term Trends**

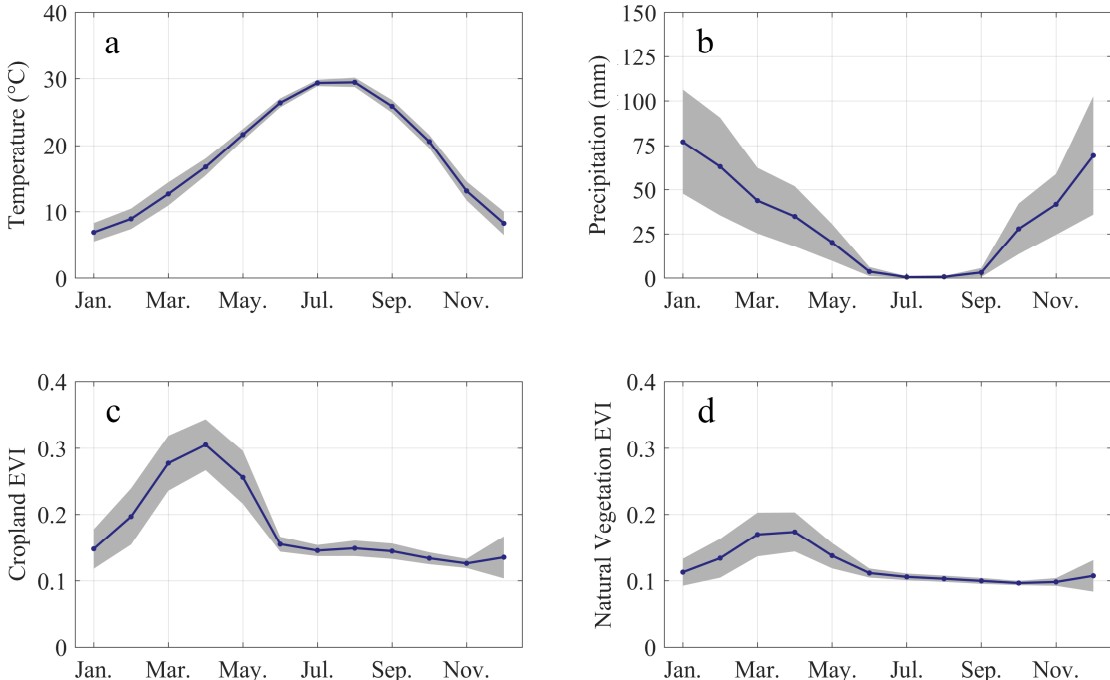

**Figure 2. monthly variations of climatic factors and vegetation of Syria. (a) – (b) Precipitation (mm) and temperature (℃). (c) – (d) EVIs of natural vegetation and croplands. All the data are the averages during 2001–2018.**

Figure 2 shows the monthly variations of the multi-year averaged hydrometeorological conditions (precipitation and temperature) and vegetation conditions in Syria. During the hot, dry summer in this region, there is little precipitation from June to September, with the total precipitation being only about 8.2 mm during this period. Rainfall begins to increase in October, reaching its peak in January, and then decreases again month by month. The monthly average temperature in winter is lower than 10℃. Since October, although the dry season has gradually turned to the rainy season, the vegetation index is

still very low, including both cropland vegetation and natural vegetation (Figure 2(c) – (d)). The growing period of cropland and natural vegetation is concentrated in February to May, and the growth of cropland vegetation is obviously better than that of natural vegetation, which is reflected in higher EVI. Therefore, the period from February to May is generally selected as the growing period.

The inter-annual variations of precipitation, temperature and vegetation are further analyzed in Figure 3. It is found that in the

165 whole year or in the growing period, both the cropland vegetation and the natural vegetation show a decreasing trend. Meanwhile, there is an increasing trend of temperature and a decreasing trend of precipitation. Therefore, intuitively, increases

in temperature and decreases in precipitation enhance water constraints, leading to further deterioration of hydrometeorological conditions in this region.

The monthly variation of climatic factors and vegetation are consistent is the inter-annual variation. Particularly, the temperature only slightly decreases in October while it increases in other months. The most obvious decrease of precipitation occurs in February (Figure 4(b)), with a rate of $-2.14$ mm yr$^{-1}$. However, vegetation in April and May decreases most significantly, including both cropland vegetation and natural vegetation. This can be most likely attributed to the precipitation decline in spring, especially the February.

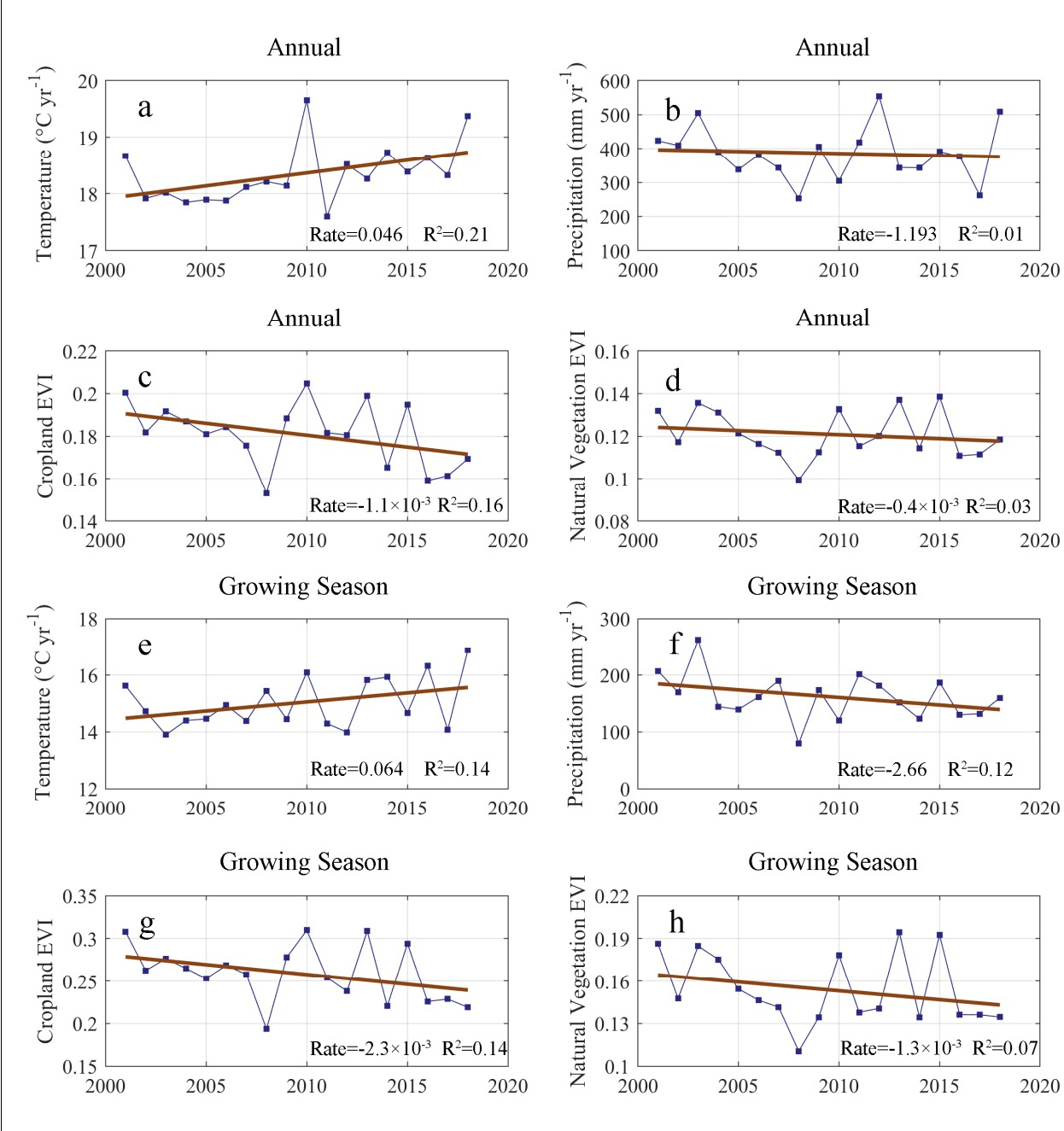

Figure 3. inter-annual variations of climatic factors and vegetation during 2001–2018. (a) – (d) are annual averages of temperature, precipitation, cropland EVI and natural vegetation EVI. And (e) – (h) are growing season (F-M) averages of temperature, precipitation, cropland EVI and natural vegetation EVI

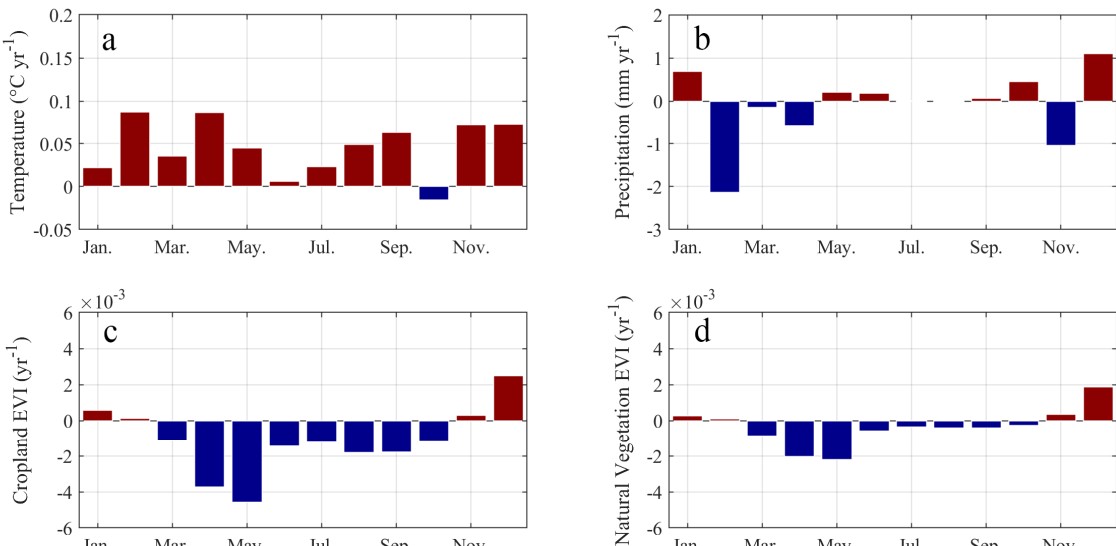

**Figure 4. inter-annual trends during 2001–2018 for each month of (a) temperature, (b) precipitation, (c) cropland EVI and (d) natural vegetation EVI.**

### 3.2 Spatial Patterns of Long-term Trends

There are large differences in the spatial variation of the vegetation in the study area. As shown in Figure 5, the EVI in the growing period shows an increasing trend in the western area, while the decreasing trend is mainly concentrated in the north

(northwest and northeast), especially for the cropland vegetation. The greening area in the south is also associated with decreases in precipitation and increases in temperature, which presents a problem in explaining this spatial distribution pattern through changes of temperature and water. Thus, it is necessary to further analyze this spatial pattern by residual analysis.

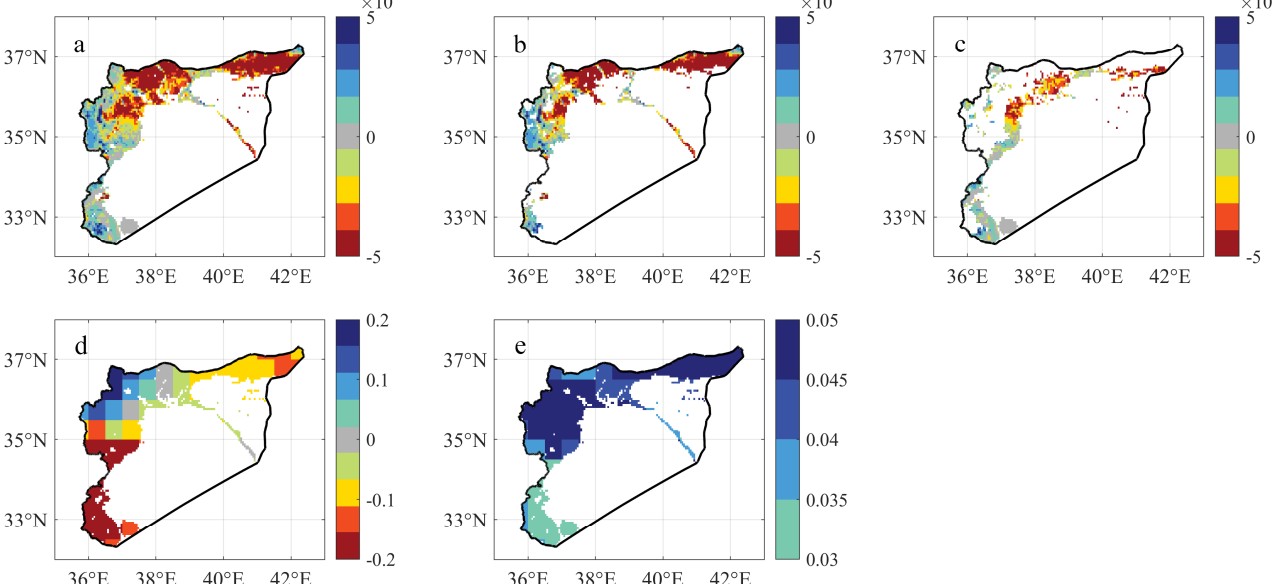

Figure 5. spatial patterns of inter-annual trends during 2001–2018 for (a) EVI in the growing period, (b) EVI of croplands, (c) EVI of natural vegetation, (d) precipitation and (e) temperature.

The vegetation change in the study area is mainly constrained by water availability. Therefore, the residual trend after removing the influence of precipitation is analyzed in this paper. Compared with overall precipitation, most of the water used by vegetation is soil moisture in root layer, which experiences cumulative and hysteresis effects (Chen et al., 2014). It is found that the EVI in the growing period is most sensitive to the cumulative precipitation from November of the previous year to April of this year (N-A precipitation for short). As shown in Figure 6, for most regions there are significant positive correlations between the N-A precipitation and the EVI in the growing period. Figure 6(b) shows the corresponding trends of residual changes. The significantly decreasing trends of the EVI residuals are mainly concentrated in northern Syria. The RUE here is expressed by the ratio of the EVI in the growing period to the N-A precipitation, of which the interannual variations also could be used to induce human activity impacts. The result shows that the RUE decreases in the north and increases in the southwest (Figure 6(c)), which is similar to that of residual analysis in spatial distribution. Therefore, based on the We suggest that in addition to climatic factors, the most important reason for vegetation decline in this area are changes in human activities.

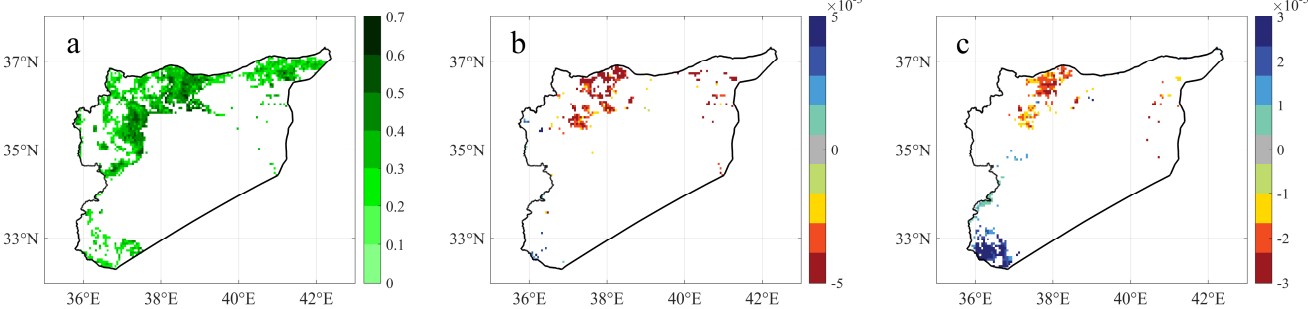

**Figure 6. (a) correlations between the N-A precipitation and the EVI in the growing period; (b) trends of EVI residuals after removing the linear fitting by precipitation (10⁻³ yr⁻¹); (c) rainfall use efficiency.**

### 3.3 Paired Land Use Experiment

In order to illustrate the land management rather than climate impact on vegetation trend, through the remote sensing data, a paired analysis has been further conducted in the agricultural area adjacent to Turkey (Figure 7). A parallelogram-shaped region located in the Khabur River Basin is selected, whose vertex coordinates are [37.5°N, 40.5°E], [36.75°N, 41°E], [37.25°N, 39.5°E] and [36.5°N, 40°E] (a keyhole markup language (KML) file is attached in the supplementary data file). This region includes parts of Turkey and Syria, where irrigated cropland is widely distributed (Hole, 2009).

As shown in Figure 7, the correlation between EVI series of which the linear trends are removed is 0.8027 (p < 0.001), suggesting both areas are affected by similar, consistent climate variations. However, the vegetation exhibits contrasting trends of greening and browning in these two regions. The inter-annual variation trends of the EVI in Syria and Turkey sides in this basin are $-3\times10^{-3}$ and $4.5\times10^{-3}$, respectively. Similarly, the seasonal distributions of the variation trend also show significant differences. Therefore, it is likely that the effect of land management contributes to this difference. One possible reason is that with the increasing in agriculture input in Turkey, the land management has promoted the growth of cropland vegetation. The other reason maybe that, the social unrest in Syria has had a serious negative impact on agricultural production, and thus the land management has had a negative effect on the growth of cropland crops.

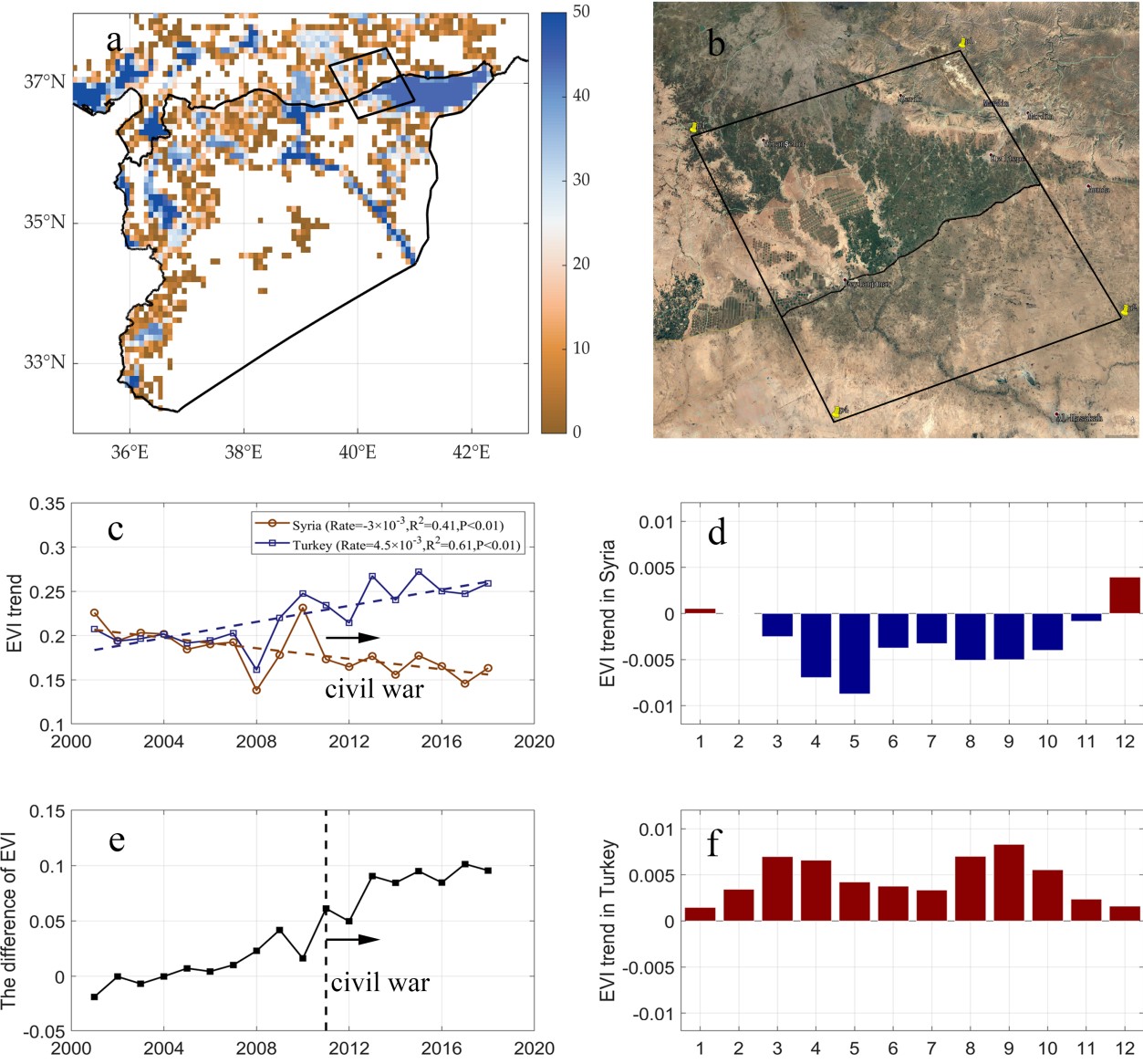

**Figure 7. (a) The percentage of the total cropland area used for irrigation for the reference year of 2005. (b) The Khabur River Basin and cropland distribution (from Google Earth © Google Earth 2019, a KML file is attached). The vertex coordinates of the parallelogram are [37.5°N, 40.5°E], [36.75°N, 41°E], [37.25°N, 39.5°E] and [36.5°N, 40°E]. (c) Annual EVI series of cropland in the Syria side and Turkey side with linear fitting. (d) EVI trends of each month of cropland in the Syria side. (e) EVI difference between Turkey and Syria sides of the basin (Turkey's EVI minus Syria's EVI). (f) EVI trends of each month of cropland in the Turkey side.**

**4. Discussion**

Recent vegetation changes have raised widespread concern. On one hand, vegetation is an important resource and ecological indicator. On the other, it also has an important feedback effect on climate, affecting the energy and water cycle of the earth-atmosphere system (Ryu et al., 2019; Xiao et al., 2019). Climatic driving factors for vegetation changes generally include temperature, precipitation and the fertilization effect of CO2. Besides, the land-use change could also cause the change of vegetation. However, there is a serious lack of the studies on land management other than land-use changes (Prestele et al., 2017).

For the study area in this paper, due to the typical arid and semi-arid climate in this region, the primary constraint for vegetation growth is the water, therefore precipitation plays a dominant role here. However, the role of land management in this area is also worthy of attention. On the one hand, irrigated agriculture is widely distributed in this area (Figure 7(a)), indicating high-intensity land management in this area. On the other hand, the economic fluctuation and social unrest in the study area are relatively large. Among them, the most typical case is the fluctuation of the investment in land management caused by social unrest, including irrigation facilities, seeds, fertilizers, pesticides, mechanization, effective field management and so on (Mason, 2019; Mohammed et al., 2020).

Therefore, it is necessary to study the inter-annual fluctuation and long-term variation trend of vegetation in this area. Because the study area is a typical water-limited ecosystem, the fluctuation of water conditions in this area will still have a significant impact on cropland vegetation and natural vegetation even if there are irrigation facilities. For example, severe droughts (Kelley et al., 2015) greatly affect the EVI during 2007–2009 (Figure 3). The average EVI of cropland vegetation is 0.24 in 2001–2018, while it is 0.22 in 2007–2009. In contrast, the average EVI of natural vegetation is 0.15 in 2001–2018, while it is 0.12 in 2007–2009. Related research reports the vegetation damage caused by this drought was mainly due to the shortage of irrigation water resources (Kelley et al., 2015; Châtel, 2014). Syrian farmers depended heavily on the dam reservoir and the extraction of groundwater. However, as early as the early 20th century, Syria's groundwater resources were already severely inadequate and land degraded. When the drought occurred in 2006 and 2008, the government's mismanagement and the already bad situation caused serious social impact. Crop failure has triggered the migration of people from rural to urban areas. In the suburbs of major cities in Syria, including Aleppo, Damascus, Deir ez-Zor, Hama and Homs, there are more than 1.5 million people migrating from rural areas to cities and refugee camps, most of whom are agricultural workers and family farmers (Eklund and Thompson, 2017; Gleick, 2014).

The long-term trend of vegetation, especially its spatial distribution, regardless of the residual analysis method or the rainfall use efficiency, indicates that the vegetation decline in the Northern Syria is caused by human activities. This is naturally linked to the devastating effects of the Syrian civil war in the region. After 2011, the EVI of the region continued to decline. In 2018, the EVI of the growing season was only 0.22, which was the lowest year except 2008. According to the report of FAO (FAO, 2020), the survey data of field research has revealed the comprehensive degradation of land management brought by war, including the insufficiency of irrigation facilities, the lack of seed, fertilizers and pesticides, as well as the inadequacy of field

management. Therefore, several research institutions demonstrated the map of social unrest during the Syria civil war since 2011. For instance, the Armed Conflict Location and Event Data Project (ACLED) records the time and location information of unrest events, including (not limiting to) the battle, pretests and violence (Raleigh et al., 2010). The Carter Center documents the territorial changes of Syria at the community level since 2014 (Carter Center, 2020). Over a certainty region, the number of control armed groups could also reflect the handover frequency and the degree of social unrest. These maps also have demonstrated that the browning areas found here have experienced serious social unrest generally. Meanwhile, the social unrest reduced land management change could be ascertained from well-documented reports in Syria, however, how to evaluate this issue on a globe scale, including the area, severity and timing, is still a huge challenge.

In a local region where the climatic conditions (including average and variations) are quite similar, vegetation changes induced by natural factors should theoretically be similar. The difference in vegetation change of the two parts in such a region could be caused by human activities rather than natural factors. The paired analysis across the border between Syria and Turkey confirmed this hypothesis. After removing the linear trend in EVI of both sides, the correlation coefficient reached 0.8. However, the annual and monthly trends of the two are completely opposite. This difference can only be attributed to human activities which refer land management in this case. In Turkey, since 1977, the government has built 22 dams, 19 hydropower stations and some irrigation networks, which has promoted the development of agriculture in the region (Eklund et al., 2017). For Syria, this can only be attributed to the adverse effects of cropland abandonment and insufficient planting caused by social unrest. Especially after the outbreak of the war in 2011, the gap of these two regions in the annual average EVI gradually increased, reaching 0.1 at the end of $2001 - 2018$.

Regional browning issues have also been discovered in a general greening world, which are mainly contributed to climate constrains, and the influences from insufficient land management has not been well identified yet (Pan et al., 2018; Yuan et al., 2019). Although this article identifies the contribution of social unrest caused by land management to regional browning, it still cannot set a certain year as a turning point (such as 2011). Both data length and severe climate variations would limit the statistical significance in turning point identification. Furthermore, even before the war started in 2011, the ability of disaster prevention and mitigation is quite poor. Once natural disasters and social unrest are superimposed, a continuous decline in land management capacity can be seen.

How to quantify the contributions of climate change and human activities in vegetation change is still challenging. In this paper, we identify the driving factor of land management on the browning based on the coherent results from multiple statistical methods. However, there is still a gap from "identify" to "quantify" the contributions.

## 5. Conclusions

In this study, we investigated vegetation changes in Syria and their potential drivers. During 2001–2018, the EVIs of the whole year and the growing period in the whole country decrease at the rates of $-0.8 \times 10^{-3}$ $yr^{-1}$ ($p < 0.1$) and $-1.9 \times 10^{-3}$ $yr^{-1}$ ($p < 0.05$), respectively. Spatial patterns indicate that the EVI decline is mainly concentrated in the northern region. Both the

residual analysis and the RUE analysis suggest that besides climate change, human activities play a significant role in the vegetation decline. A paired analysis in the Khabur River Basin across the border of Syria and Turkey demonstrates that both sides of the region have widely-distributed croplands. EVIs on two sides have similar trend-removed inter-annual (climatic) variations but contrary long-term trends, and the Syria side is browning while the Turkey side is greening. The spatial patterns between the browning areas and social unrest of the civil war are generally coherent and relative reports have reported that social unrest had induced insufficient irrigation and lack of seeds, fertilizers, pesticides and field managements. These changes can be classified into the category of land management. Therefore, we concluded that the decline of vegetation condition (browning) in the north Syria is driven by the change of land management rather than climate change.

*Acknowledgments.* We thank Nanjing Hurricane Translation for reviewing the English language quality of this paper, and thank Dr. Xiaogang He for the discussion of this paper. This research is supported by the National Natural Science Foundation of China (NO. 42130506, 31570464), the National Key R&D Program of China (NO. 2017YFB0504000). CRU TS4.03 is available at the Centre for Environmental Data Analysis (http://browse.ceda.ac.uk/). EVI and Land cover type products were obtained from the LAADS DAAC (Level–1and Atmosphere Archive & Distribution System Distributed Active Archive Center https://ladsweb.modaps.eosdis.nasa.gov).

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
