# Peer review of "Land Management Contributes significantly to observed Vegetation Browning in Syria during 2001–2018"

_Biogeosciences, 2021_

## Author Comment (AC1)

***Comments from the Reviewer:***

***Reviewer #1 (Formal Review for Authors):***

*Since the remote sensing vegetation indices have been well developed, extensive research has been conducted on vegetation change (greening or browning) and corresponding driver identification. Studies about the impact of land management is still lacking due to data and methods limitations. In this paper, the authors demonstrated a typical browning case over Syria by involving land management due to social unrest, which is quite innovative. This paper fits the scope of Biogeosciences and uses several methods with coherent results to support the conclusion. Therefore, an acceptance is suggested with some minor comments.*

Dear Reviewer,

Thanks for your time. We strive to present an original work and your suggestion are of great help to improve this paper.

Regards,

Prof. Dr. Tiexi Chen and coauthors

*1) In section 2.2, the authors need to clarify which year or which period of the land cover data are used to abstract different vegetation types.*

**REPLY**: Thanks for your comments. The land cover data was used in 2010 which is in the middle of the study period, this detail has been added in the revised version.

*2) cropland or farmland? Should be consistent*

**REPLY**: Thanks for your comments. We checked the full text and unified it as cropland.

*3) Line 171, "The vegetation change in the study area is mainly constrained by the soil moisture and precipitation." this sentence is ambiguous, soil moisture and precipitation are not independent factors.*

**REPLY**: Thanks for your comments. We agree with your concern. A precise term should be water availability, which could be indicated by precipitation, soil moisture and other related indices. Therefore, in the revision, this sentence is:

"The vegetation change in the study area is mainly constrained by water availability."

*4)  The growing period is firstly defined in 3.1 section, which is used in the following analysis. The growing season usually defined in the method section. Meanwhile, growing season also should be labeled in figure 3.*

**REPLY**: Thanks for your comments. The use of growing season is presented in the section 2.6 as "The growing season is defined from February to May (F-M) based on the vegetation phenology (Figure 2)." Meanwhile, both "Annual" and "Growing Season" were added as titles in Figure 3.

[Figure]

**Figure 3. inter-annual variations of climatic factors and vegetation during 2001–2018. (a) – (d) are annual averages of temperature, precipitation, cropland EVI and natural vegetation EVI. And (e) – (h) are growing season (F-M) averages of temperature, precipitation, cropland EVI and natural vegetation EVI**

*5)   irrigation distribution data was used in this study which is not mentioned in the method section?*

**REPLY**: Thanks for your comments. The irrigation information was added as a new section 2.5 being "2.5 Cropland irrigation data".

2.5 Cropland irrigation data

Irrigation is an important method for conducting farmland management and increasing production, especially for dryland. Irrigation facilities are also vulnerable to severe social stability and economic fluctuations. We selected the "Global Irrigation District Map" (latest version 5) from FAO (Food and Agriculture Organization of the United Nations), which shows the percentage of the total cropland area used for irrigation around a reference year 2005 with a resolution of 5 minutes (Stefan et al., 2013).

*6)   section 3.3 is quite interesting, Figure 7b is not clear, maybe the authors could demonstrate the subregion in Figure 7a.*

**REPLY**: Thanks for your comments. Due to the color problem, the rectangle and border line in Figure 7b are blurry, we adjusted the color with black, and the figure is clearer now, as shown below. We also added this box in Figure 7a.

[Figure]

**Figure 7. (a) The percentage of the total cropland area used for irrigation for the reference year of 2005. (b) The Khabur River Basin and cropland distribution (from Google Earth © Google Earth 2019, a KML file is attached). The vertex coordinates of the parallelogram are [37.5°N, 40.5°E], [36.75°N, 41°E], [37.25°N, 39.5°E] and [36.5°N, 40°E]. (c) Annual EVI series of cropland in the Syria side and Turkey side with linear fitting. (d) EVI trends of each month of cropland in the Syria side. (e) EVI difference between Turkey and Syria sides (Turkey's EVI minus Syria's EVI). (f) EVI trends of each month of cropland in the Turkey side.**

*7) in the discussion section, the significance of Figure 7 is not well illustrated.*

**REPLY**: Thanks for your comments. We also realize that the content shown in Figure 7 is a very meaningful result, and the idea is worth digging deeper. In the revised version, we added a paragraph in the discussion section:

"In a local region where the climatic conditions (including average and variations) are quite similar, vegetation changes induced by natural factors should theoretically be similar. The difference in vegetation change of the two parts in such a region could be

caused by human activities rather than natural factors. The paired analysis across the border between Syria and Turkey confirmed hypothesis. After removing the linear trend in EVI of both sides, the correlation coefficient reached 0.8. However, the annual and monthly trends of the two are completely opposite. This difference can only be attributed to human activities which refer land management in this case. In Turkey, since 1977, the government has built 22 dams, 19 hydropower stations and some irrigation networks, which has promoted the development of agriculture in the region (Eklund et al., 2017). For Syria, this can only be attributed to the adverse effects of cropland abandonment and insufficient planting caused by social unrest. Especially after the outbreak of the war in 2011, the gap of these two regions in the annual average EVI gradually increased, reaching more than 0.1 at the end of 2001 – 2018."

---

## Author Comment (AC2)

*Comments from the Editor:*

*While most scientific studies on terrestrial vegetation cover changes and their effects on the climate system, were focused on human activities of deforestation/afforestation and managements changes in agricultural and in pasturing areas, social unrest and human conflicts effects on the land cover are poorly studied. This paper is among the first to assess this interesting topic, providing important information and research approach that will be relevant also for wide scientific communities and clear fits the BG journal agenda. In the followings the authors will find comments and suggestions to their manuscript.*

Dear reviewer,

Thanks for your time and comments. We strive to present an innovative work and your suggestion are of great help to improve this paper.

Regards,

Prof. Dr. Tiexi Chen and coauthors

*General comments:*

*1)      The authors rightly mentioned that Syrian conflict is an example for others, nations wars or regional unrests which unfortunately persisting with no end, and are also effecting the land covers around the world. Thus, adding estimation information on a global size of areas affected by such conflicts over recent years will provide demonstration of this unaware aspect and its importance for the regional climate system and for land surface ecological services of those areas.*

**REPLY**: Thanks for your comments. This is a very nice question. During the discussion periods, the authors also want to quantify the conflicts but we still do not know how to make this work. As mentioned in the discussion, we have found the data sources from the Armed Conflict Location and Event Data Project (ACLED).

We totally agree with your opinion, and at current stage, at least we could give a qualitative description. Therefore, in the introduction, the followings are added:

"Social unrest is still a global problem, which is bound to affect regional development and land management activities. However, social unrest has been rarely mentioned in the study of land use and management. Based on the Armed Conflict Location and

Event Data Project (ACLED) records (Raleigh et al., 2010), in the past 20 years, there are about 20 countries that have experienced more than 1000 battles. Several countries have involved in national wars, such as (not limited to) Syria, Libya, Afghanistan, Iraq, Yemen. Therefore, the affected population and land area are at least in the order of 100 million and tens of millions of square kilometers, respectively."

In the discussion section, we added:

"Meanwhile, the social unrest reduced land management change could be quantified based on well-documented reports in Syria, however, how to evaluate this issue on a globe scale, including the area, severity and timing, is still a huge challenge."

*2) Syria is a large country with different climatic regions and its great part is a desert land. Much of the conflicts are concentrated in the northern regions and around the capital. However, the information provided (e.g., figure 2 & 3) is for the whole country, likely not well representing the climatic conditions in the main conflict zones.*

REPLY: Thanks for your comments. We agree with this and in fact, our specific analysis only includes the vegetated area, that is, land cover type 1 and 2 in Figure 1(a). The temperature, precipitation and EVI in Figures 2 and 3 are only for vegetation areas, not including deserts, water and built-up lands. We added related descriptions in section 2.1. From Figure 5 you can see that no-vegetated areas are not included in our analysis.

*3) Separating the climate from human impacts on the land cover changes over the years is at the heart of the paper; a figure and deeper discussion presenting and explaining the EVI decline due to either of the two is missing.*
*While the conflict effect in the Khabur River region is a good example, the authors may find another such region along the Turkey border that has not experienced conflict to emphasize that the effect is due to unrest.*

REPLY: Thanks for your comments. Indeed, you are rising the essential question which currently plagues the entire academic community. How to quantify the contributions of climate change and human activities in vegetation change? This problem has been torturing me since the very beginning of my career. Generally, there

are two ways: Statistical methods based on observations and comparative analysis of scenarios based on models. Both of these methods are not satisfactory. Models are easy to quantify but are powerless for processes that cannot be effectively described, including land management. The assumptions of observation-based residual analysis are difficult to satisfy in real situations, although this method is also applied here in this article.

Therefore, I divided the problem into two aspects: identifying contributions and quantifying contributions. In this paper, several statistical methods are used instead of models. At the same time, we do not dare to believe in the results of one single method, but find coherent conclusions through multiple methods. Paired analysis is a quite new idea, and we are still trying to systematically explain it. As you suggested, in the revision more discussions are added in section 2.6:

"In this paper, we applied statistical methods on observations rather than using models. Coherent results of these method are expected to lead a more robust conclusion."

"A paired analysis over two parts of a local region is also applied. Both parts have identical climate variations and the difference in vegetation changes of these two parts could be contributed to human activities rather than climate drivers."

"How to quantify the contributions of climate change and human activities in vegetation change is still challenging. In this paper, we identify the driving factor of land management on the browning based on the coherent results from multiple statistical methods. However, there is still a gap from "identify" to "quantify" the contributions."

As you suggested, the following figure shows the spatial trend of all cropland on the border between Turkey and Syria during the growing season. The cross-border plain area is an ideal study area, with relatively extensive farmland distribution on both sides. Therefore, this study area is the only typical case we have found on the border between the two sides.

[Figure]

Figure R1, The spatial variation trend of EVI in the growing season from 2001 to 2018

*Specific comments:*

*1.     Sentences starting in line 98. Dryland areas characterized by a low, relative spares vegetation cover, so what is the advantage of using EVI over NDVI? Did the authors check this? This may provide similar results but may the opposite? Obviously, this cannot be checked on the ground, however it can be analyzed with e.g., google RGB products.*

**REPLY**: Thanks for your comments. As you mentioned, EVI is usually applicable to areas with dense vegetation. In the arid area of Syria, the effects of EVI and NDVI on characterizing vegetation should be similar. To make this point clear, we further applied NDVI (same sources of MODIS) in this region. Both NDVI and EVI have consistent results (Figure r2,

r3).

[Figure]

Figure r2. (a) correlations between the N-A precipitation and the NDVI in the growing period; (b) trends of NDVI residuals after removing the linear fitting by precipitation; (c) rainfall use efficiency.

[Figure]

Figure r3. The annual trend of NDVI during 2001- 2018

2.      *L 114. The meaning of "…improved spatial consistency…", or provide reference to this model.*

**REPLY**: Thanks for your comments. This is a clerical error. What we want to express is that the meteorological environment is more uniform in space than the surface parameters. Therefore, in areas with relatively simple topography, the variations in temperature and precipitation within a 0.5 degree resolution is very small. In the revision, we deleted "improved". Now the sentence is:

"Due to spatial consistencies of temperature and precipitation over small regions, it is assumed that the vegetation within the 0.5° interval has the same environmental condition."

*3.    Provide reference to the sentence starting in L. 122.*

**REPLY**: Thanks for your comments. We have added two references in the revised version.

Dardel, C. c., Kergoat, L., Hiernaux, P., Grippa, M., Mougin, E., Ciais, P., & Nguyen, C.-C.. Rain-Use-Efficiency: What it tells us about the Conflicting Sahel Greening and Sahelian Paradox. Remote Sens., 6(4), 3446-3474. https://doi.org/10.3390/RS6043446, 2014.

Fensholt, R., Rasmussen, K., Kaspersen, P. S., Huber, S., Horion, S., & Swinnen, E.. Assessing Land Degradation/Recovery in the African Sahel from Long-Term Earth Observation Based Primary Productivity and Precipitation Relationships. Remote Sens., 5(2), 664-686. https://doi.org/10.3390/RS5020664, 2013.

*4.    Comment to the representativeness of Figure 2 & 3 has already mentioned. Specially to crops and to natural vegetation it is suggest to concentrate in the conflict zones. By the way, 'natural vegetation' included different vegetation types (as authors mentioned) and they have slightly different seasonality patterns…*

**REPLY**: Thanks for your comments. We agree with this and as reply above, our specific analysis only includes the vegetated area, that is, land cover type 1 and 2 in Figure 1(a). The temperature, precipitation and EVI in Figures 2 and 3 are only for vegetation areas, not including deserts, water and built-up lands.

Regarding the conflict area, we only know the general scope, and we are still thinking about how to specifically divide the space, including the issue you mentioned above, how to map this issue globally.

Syria's natural vegetation composition structure is relatively simple, of which about 80.0% are open shrublands, grassland accounts for 18.1, and the other accounts for less than 2%.

*5.    Which years figures 5 and 6 are covering?*

**REPLY**: Thanks for your comments. All the pictures are for 2001-2018, we have added the time periods in the figure caption.
*Syria civil war started in May 2011, while EVI's declines, according to Figure 3,*

*started already before. Can the authors separate the 2 periods to show the human influence?*

**REPLY**: Thanks for your comments. This was also our expectation at the very beginning. In the subsequent analysis, we found it difficult to set the turning point. Potential reasons area: 1, the data length is not long (2011-2018 only have 8 years) and climate variations are very severe, such as the drought in 2008. Therefore, Due to insufficient sample size and large fluctuations, it is difficult to obtain statistically significant turning point. 2, although war is an emergent event, its destruction is also a gradual process. And before the war broke out, there were already signs of social unrest in the area, and under the influence of severe natural disasters (droughts), an unstable society would make disaster prevention and mitigation inefficient. When the two are superimposed, a continuous decline in land management capacity can be seen.

"Regional browning issues have also been discovered in a general greening world, which are mainly contributed to climate constrains, and the influences from insufficient land management has not been well identified yet (Pan et al., 2018; Yuan et al., 2019). Although this article identifies the contribution of social unrest caused by land management to regional browning, it still cannot set a certain year as a turning point (such as 2011). Both data length and severe climate variations would limit the statistical significance in turning point identification. Furthermore, even before the war started in 2011, the ability of disaster prevention and mitigation is quite poor. Once natural disasters and social unrest are superimposed, a continuous decline in land management capacity can be seen."

6.     *Figure 7: Explain the meanings and provide the units (of 'irrigation distribution'). b. can the border line (Turkey to Syria) be added to the picture and if possible also adding picture from a period before the civil war, with their dates. c-f. Make clear that EVI is of the irrigated area only.*

**REPLY:** Thanks for your comments. (a) The description of irrigation data is added in section 2.5, and the unit is also explained under figure 7 caption. (b) Corresponding modifications were made in Figure 7. c-f is only the trend of the cropland within the rectangle, which has also been modified in the labels and caption.

[Figure]

**Figure 7. (a) The percentage of the total cropland area used for irrigation for the reference year of 2005. (b) The Khabur River Basin and cropland distribution (from Google Earth © Google Earth 2019, a KML file is attached). The vertex coordinates of the parallelogram are [37.5°N, 40.5°E], [36.75°N, 41°E], [37.25°N, 39.5°E] and [36.5°N, 40°E]. (c) Annual EVI series of cropland in the Syria side and Turkey side with linear fitting. (d) EVI trends of each month of cropland in the Syria side. (e) EVI difference between Turkey and Syria sides (Turkey's EVI minus Syria's EVI). (f) EVI trends of each month of cropland in the Turkey side.**

7.   *L 230. Fig. 3 shows EVI decline prior to the civil war and even prior to 2007-09 drought. It possible to claim that the EVI decline was due to mismanagement leading to this civil unrest but it needs to provide evidences for that.*

**REPLY**: Thanks for your comments. Both the literature survey and the relevant data show that Syria is a country highly dependent on irrigation, and the water resources for the irrigation mainly come from groundwater. At the beginning of the 21st century, the local water resources were already so tense that the subsequent drought

exacerbated the situation of poor agricultural harvests. We expanded the discussion section to better illustrate that land management in the area was insufficient before the war began. Please also refers to the replies to comment 5 above.

"Related research reports the vegetation damage caused by this drought was mainly due to the shortage of irrigation water resources (Kelley et al., 2015; Châtel, 2014). Syrian farmers depended heavily on the dam reservoir and the extraction of groundwater. However, as early as the early 20th century, Syria's groundwater resources were already severely inadequate and land degraded. When the drought occurred in 2006 and 2008, the government's mismanagement and the already bad situation caused serious social impact."

*8.      Sentence in L 235 is unclear.*

**REPLY**: Thanks for your comments. This refers to some research institutions, we have corrected this sentence in revised version.

"Therefore, several research institutions demonstrated the map of social unrest during the Syria civil war since 2011."

*9.      247. Based on what is the claim that if removing the human activity EVI change will be the same for both sides of the border?*

**REPLY**: Thanks for your comments. We agree and notice this part should be explained and discussed more in details. In the revision, more discussions were made about the pared analysis.

 "In a local region where the climatic conditions (including average and variations) are quite similar, vegetation changes induced by natural factors should theoretically be similar. The difference in vegetation change of the two parts in such a region could be caused by human activities rather than natural factors. The paired analysis across the border between Syria and Turkey confirmed this hypothesis. After removing the linear trend in EVI of both sides, the correlation coefficient reached 0.8. However, the annual and monthly trends of the two are completely opposite. This difference can only be attributed to human activities which refer land management in this case. In Turkey, since 1977, the government has built 22 dams, 19 hydropower stations and some irrigation networks, which has promoted the development of agriculture in the region (Eklund et al., 2017). For Syria, this can only be attributed to the adverse

effects of cropland abandonment and insufficient planting caused by social unrest. Especially after the outbreak of the war in 2011, the gap of these two regions in the annual average EVI gradually increased, reaching more than 0.1 at the end of 2001 – 2018."

10.     *Figure 7. Why EVI increases so much in the Turkey side?*
*Additionally, EVI trends deviation between the two countries has started already in the early 2000ths, even before the 2007 drought, how this explained?*

**REPLY**: Thanks for your comments. At present, global greening is a widespread phenomenon, especially for agricultural areas. Many factors have contributed to global greening, including $CO_2$ fertilization, nitrogen deposition, climate change, land management, etc (Chen, et al. 2019;Zhu, et al. 2016). In Turkey, on the one hand, the effect of $CO_2$ fertilization has led to the growth of cropland EVI. On the other hand, land management has made significant contributions. Since 1977, the government has successively built 22 dams, 19 hydroelectric power plants and some irrigation networks, which promotes the development of agriculture in the region (Eklund, et al. 2017).

Chen, Chi, et al. "China and India Lead in Greening of the World through Land-Use Management." Nature Sustainability, vol. 2, no. 2, 2019, pp. 122–129.

Zhu, Zaichun, et al. "Greening of the Earth and Its Drivers." Nature Climate Change, vol. 6, no. 8, 2016, pp. 791–795.

Eklund, Lina, and Darcy Thompson. "Differences in Resource Management Affects Drought Vulnerability across the Borders between Iraq, Syria, and Turkey." Ecology and Society, vol. 22, no. 4, 2017, p. 9.

As shown in the figure below, before 2007, the EVI gap between Turkey and Syria on the border of cropland was small, while the gap gradually increased during the drought. After the outbreak of the war in 2011, the gap between the two reached more than 0.1, that is to say, this difference in two trends is mainly due to insufficient land management caused by the Syrian war. Figure 7 (c) also shows this

difference.

[Figure]

Figure r4, annual EVI time series of the croplands in The Khabur River Basin over Turkey (red) and Syria (blue) sides.

---

## Author Response (AR1)

Comments from editor:

Dear Authors,

Thanks for proceeding with us with your paper.

Going over your replies to the reviewers comments I see the following as a major weakness of the paper:

The new interesting and important text you added emphasized that unlike Syria, Turkey invested much in irrigation and water supply to the agricultural sector, although it is unclear if this included the Khabur basin region? Figure 7.e shows the gap between the two countries already started in 2003 and increases to high values already before the war (2011). It may raise a more general point for you to consider: that the underinvestment in irrigation in the Syrian agricultural sector led to the unrest and the war. It will be interesting if this can be shown using the RS technique. As Khabur region is a small fraction of the Syrian agricultural land, I would suggest to expand this to larger parts of the country and back it up with independent information on irrigated land/water amount changes in Syria before the war.

Note please, according to Fig. 7.a, the larger and massive cropland area (dark blue) in the Syrian side is in the right side of the study area. When comparing fig' 7.a. (2005) with b. (2019), it is clear that the drastic land cover change is at that same area. Maybe, you need to concentrate the changes in this blue region?

According to Fig. 7.b the difference there is very dramatic, unlike the data in fig' 7.c.

Fig's d & E are suggesting that the image (Fig b) is from the summer (is it right?) and thus hint to lack of irrigation due to the unrest – isn't it?

Looking forward for the new version.

Hope this find you in a good health. Kind regards, Eyal

**REPLY:** Dear Editor, thank you very much for your in-depth thinking, which has given us a lot of inspiration.

1. Indeed, we do not know the details in Khabur basin of the Turkey side, and we only can assume that in a stable country, the general trends (increasing or decreasing) in agriculture investments of different region are consistent. It's beyond our ability to fully understand what's going on in this area.

2. Irrigation data is difficult to obtain and there is no mature technology yet (Meier et al., 2018). A recent released irrigation dataset developed by Nagaraj et al. (2021). As illustrated by Figure r1, irrigated cropland in Syria appears to have continued to decline. However, some of the recovery in 2015 compared to 2010 seems inexplicable. We also examined some areas of China with well-known irrigation changes and found some unreasonable results using this dataset. This suggests that the data extracted from changes in irrigation are currently very unreliable. Therefore, we did not use this dataset in the paper. Unfortunately, FAO based data is only updated to 2005 (as shown in fig.7a). Meanwhile, the map (picture) from google earth is for

illustrative purpose only (synthesized from different satellites at different times, Figure r2-3), which cannot be applied on the quantitative analysis.

[Figure]

Figure r1, the changes of irrigation status in Syria. Three types are defined: 0 – 2 represents no irrigation, medium irrigation and high irrigation. The changes were calculated by the differences which cover a range of -2 to 2. A positive number represents an increase in irrigation, a negative number represents a decrease in irrigation. a) the changes between 2010-2001; b) the changes between 2015-2001; c) the changes between 2015-2001.

[Figure]

Figure r2, image from google earth web of the Syria side. It says: date 2019/7/15 CNES/Airbus

[Figure]

Figure r3, image from google earth web of the Turkey side. It says: date 2019/11/11 Maxar Technologie…

3. There is a very critical assumption on the paired analysis. The two regions should have identical climate conditions (control the climate driven part), therefore, generally a divide plain is relatively ideal. As illustrated in Figure r4, the whole region is too large in length. Meanwhile, the corresponding north side is a mountainous area, therefore, a comparative study cannot be carried out. This area appears to be Kurdish-controlled, and I don't know exactly what's going on.

[Figure]

Figure r4, A map of northern Syria taken from Google Earth. The yellow line represents a distance of 217 km.

4. The emergence of social unrest is an extremely complex social problem, which is beyond the scope of this paper. Particularly, the same phenomena and evidence can be interpreted politically in diametrically opposite ways. From a scientific point of view, it can neither be experimentally confirmed nor quantified. However, the scientific logic of the impact of human activities on the ecosystem itself is clear and definite. Meanwhile, further reading is suggested with a latest paper which titled "Civil war hinders crop production and threatens food security in Syria".

I must honestly say that the analysis can only be done on a national scale at the moment. It is a pity that I am unable to actually investigate and find out the details about the area, all through news and related reports. Sorry for not fully answering your concerns.

The details of the revision are in the track change manuscript, and we are very grateful to the reviewers for their comments and the editors' work.

References:

Meier, J., Zabel, F., & Mauser, W. (2018). A global approach to estimate irrigated areas–a comparison between different data and statistics. Hydrology and Earth System Sciences, 22(2), 1119-1133.

Nagaraj, D., Proust, E., Todeschini, A., Rulli, M. C., & D'Odorico, P. (2021). A new dataset of global irrigation areas from 2001 to 2015. Advances in Water Resources, 152, 103910.

Li, X. Y., Li, X., Fan, Z., Mi, L., Kandakji, T., Song, Z., ... & Song, X. P. (2022). Civil war hinders crop production and threatens food security in Syria. Nature Food, 1-9.

---

## Author Response (AR2)

*Comments to the author:*
*Thanks Authors, for the interesting paper.*
*The degradation of groundwater reservoirs is referred (line 304) to the early 20th century, but likely it is for the early 21st century…?*
*Kind regards and good luck with your research activities, Eyal*

**Reply:**

Dear editor, thanks so much for your work and your contributions during the review process, which help us a lot.

Thanks for your comment, the "the early 20$^{th}$ century" should be "the end of the 20$^{th}$ century".

We have edited the manuscript strictly based on the submission guidelines, including figure format, and added necessary sections such as data availability ...

We are glad to have this opportunity to publish our innovative paper in Biogeosciences. We will also continue to submit important results to your journal.

Regards!

Prof. Dr. Tiexi Chen and coauthors